# OMNIGROK: GROKKING BEYOND ALGORITHMIC DATA

**Ziming Liu, Eric J. Michaud & Max Tegmark**
Department of Physics, Institute for AI and Fundamental Interactions, MIT
{zmliu,ericjm,tegmark}@mit.edu

## ABSTRACT

Grokking, the unusual phenomenon for algorithmic datasets where generalization happens long after overfitting the training data, has remained elusive. We aim to understand grokking by analyzing the loss landscapes of neural networks, identifying the dependence of the generalization gap on model weight norm as a cause of grokking. We refer to this as the "LU mechanism" because training and test losses (against model weight norm) typically resemble "L" and "U", respectively. This mechanism can explain many aspects of grokking: data size dependence, weight decay dependence, the emergence of representations, etc. Guided by the intuitive picture, we are able to induce grokking on tasks involving images, language and molecules, although the grokking signals are sometimes less dramatic. We attribute the dramatic nature of grokking for algorithmic datasets to representation learning.

## 1 INTRODUCTION

Generalization lies at the heart of machine learning. A good machine learning model should arguably be able to generalize fast, and behave in a smooth/predictable way under changes of (hyper)parameters. Grokking, the phenomenon where the model generalizes long after overfitting the training set, has raised interesting questions after it was observed on algorithmic datasets by Power et al. (2022):

**Q1** *The origin of grokking*: Why is generalization much delayed after overfitting?

**Q2** *The prevalence of grokking*: Can grokking occur on datasets other than algorithmic datasets?

This paper aims to answer these questions by analyzing neural loss landscapes:

**A1** Grokking can result from a mismatch between training and test loss against model weight norm. Specifically, (reduced) training and test losses plotted against model weight norm resemble "L" and "U", respectively, as shown in Figure 1b. We refer to this phenomenon as the "LU mechanism", which we elaborate on in Section 2 and 3.

**A2** **Yes**. Indeed, we demonstrate grokking for a wide range of machine learning tasks in Section 4, including image classification, sentiment analysis and molecule property prediction. Grokking signals observed for these tasks are usually less dramatic than for algorithmic datasets, which we attribute to representation learning in Section 5.

Partial answers to **Q1** are provided in recent studies: Liu et al. (2022) attribute grokking to the slow formation of good representations, Thilak et al. (2022) attempts to link grokking to the slingshot mechanism of adaptive optimizers, and Barak et al. (2022) uses Fourier gap to describe hidden progress. This paper aims to understand grokking through the lens of neural loss landscapes. Our landscape analysis is able to explain many aspects of grokking: data size dependence, weight decay dependence, emergence of representations, etc.

The paper is organized as follows: In Section 2, we review background on generalization, and introduce the *LU mechanism*. In Section 3, we show how the LU mechanism leads to grokking for a toy teacher-student setup. In Section 4, we show that the intuition gained from the toy problem can transfer to realistic datasets (MNIST, IMDb reviews and QM9), for which we also observe grokking, although in a slightly non-standard setup where it is relatively weak. In Section 5, we discuss why grokking is more dramatic for algorithmic datasets than on others (e.g., MNIST), by comparing their loss landscapes. We review related work in Section 6 and summarize our conclusions in Section 7. Code is available at https://github.com/KindXiaoming/Omnigrok.

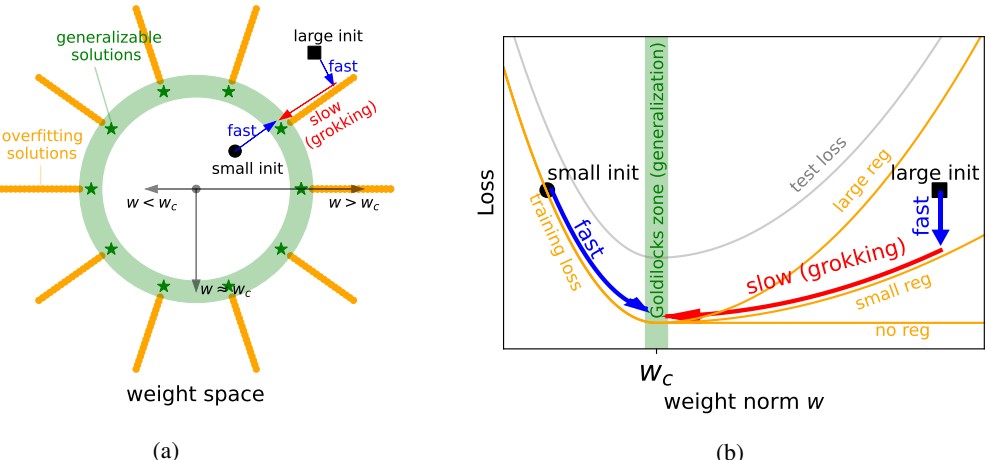

Figure 1: (a) $w$: $L_2$ norm of model weights. Generalizing solutions (green stars) are concentrated around a sphere in the weight space where $w \approx w_c$ (green). Overfitting solutions (orange) populate the $w \gtrsim w_c$ region. (b) The training loss (orange) and test loss (gray) have the shape of L and U, respectively. Their mismatch in the $w > w_c$ region leads to fast-slow dynamics, resulting in grokking.

## 2  THE LU MECHANISM FOR GROKKING

**Weight norm and reduced loss** Letting $\mathbf{w}$ denote the weights of a model, any function $f(\mathbf{w})$ (e.g, train/test loss/accuracy) depends on both the weight norm $w \equiv ||\mathbf{w}||_2$ and the angular direction $\hat{\mathbf{w}} \equiv \mathbf{w}/w$. Similar to Fort and Scherlis (2019), we define a reduced function $\tilde{f}(w)$ by minimizing training loss $l_{\text{train}}(\mathbf{w})$ over angular directions, i.e.,

$$\tilde{f}(w) \equiv f(\mathbf{w}^*(w)), \quad \text{where } \mathbf{w}^*(w) \equiv \underset{||\mathbf{w}||_2=w}{\operatorname{argmin}} \, l_{\text{train}}(\mathbf{w}). \tag{1}$$

In this paper, we set $f$ as train/test loss/error, but it also applies to other metrics of interest. In practice, we perform the constrained minimization by rescaling the model weights back to their original norm after each unconstrained optimization step. We will see that this reduced 1D loss landscape, which is easy to visualize, captures important features related to grokking. Throughout the paper, our model is initialized by multiplying a factor $\alpha \equiv w/w_0$ to the standard initialization [1], where $w_0$ and $w$ are the weight norm of the network before and after multiplying by $\alpha$, respectively.

**LU mechanism** Although the loss landscapes of neural networks are nonlinear, Fort and Scherlis (2019) reveal a simple landscape picture: There is a spherical shell in the weight space (the "Goldilocks" zone), where generalization is better than outside this zone. We illustrate the Goldilocks zone as the green area with average radius $w_c$ in Figure 1a; the green stars are the generalizing solutions. The test loss is thus higher either both when $w > w_c$ and $w < w_c$, forming a U-shape against $w$ in Figure 1b (gray curve). By contrast, the training loss has an L-shape against weight norm . There are many solutions which overfit training data for $w > w_c$, but high training losses are incurred for $w < w_c$. This corresponds to the L-shaped curve seen in Figure 1b (orange curve, no regularization). In summary, the (reduced) training loss and test loss are L-shaped and U-shaped against weight norm, respectively, which we will refer to as the LU mechanism throughout the paper.

It is well known in statistics that generalization error has a "U" shape against model capacity, which is usually attributed to the bias-variance trade-off. Although this common wisdom was challenged by the observation of double descent (Nakkiran et al., 2021), the "U" curve can be recovered from a double descent simply by changing the x-axis from the number of model parameters $N$ to the 2-norm of model parameters $w \equiv ||\mathbf{w}||_2$, at least for linear regression (Ng and Ma, 2022). Although the LU mechanism may remind readers of related phenomena (Schoenholz et al., 2016; Yang and Schoenholz, 2017; Nakkiran et al., 2021), their setups are not exactly the same as ours. More importantly, our focus and contribution is to understand grokking, a brand new generalization puzzle.

---

[1]By "standard initialization" we mean the default one in PyTorch. For linear layers, each weight $w \sim U[-\sigma, \sigma]$ and bias $b \sim U[-\sigma, \sigma]$ where $\sigma = 1/\sqrt{\text{fan\_in}}$, and $U[a, b]$ denotes uniform distribution on $[a, b]$.

**Grokking dynamics** We identify the "LU mechanism" as the cause of grokking. If the weight norm is initialized to be large (e.g., the black square in the $w > w_c$ region), the model first quickly moves to a nearby overfitting solution by minimizing the training loss. Without any regularization, the model will stay where it is, because the gradient of the training loss is almost zero along the valley of overfitting solutions, so generalization does not happen. Fortunately, there are usually explicit and/or implicit regularizations that can drive the weight vector towards the Goldilocks zone $w \approx w_c$. When the regularization magnitude is non-zero but small, the radial motion can be (arbitrarily) slow. If weight decay is the only source of regularization, and training loss is negligible after overfitting, then weight decay $\gamma$ causes $w(t) \approx \exp(-\gamma t)w_0$, when $w_0 > w_c$, so it takes time $t \approx \ln(w_0/w_c)/\gamma \propto \gamma^{-1}$ to generalize. A small $\gamma$ results in a huge generalization delay (i.e., grokking). The dependence on regularization magnitudes is illustrated in Figure 1b: no generalization at all happens for $\gamma = 0$, small $\gamma$ leads to slow generalization (grokking), and large $\gamma$ leads to faster generalization [2]. The above analysis only applies to large initializations $w > w_c$. Small initializations $w < w_c$ can always generalize fast [3], regardless of regularization.

**Why isn't grokking commonly observed?** The standard initialization schemes typically initialize $w$ no larger than $w_c$. However, if we increase initialization scales (explicitly or implicitly), grokking can appear. In Section 3 and 4, we find that explicitly increasing initialization weight norm can induce grokking. In Section 5, we argue for algorithmic datasets because (shown in Figure 6d)

$$w_c(\text{bad representation}) > w_c(\text{good representation}), \tag{2}$$

i.e., a proper initialization for a bad representation is effectively too large for a good representation, leading to grokking. Take the addition (base $p$) for example: with the good (linear) representation or a bad (random) representation, the decoder needs to learn to classify $O(p)$ or $O(p^2)$ examples, respectively.

## 3 GROKKING FOR A TEACHER-STUDENT SETUP

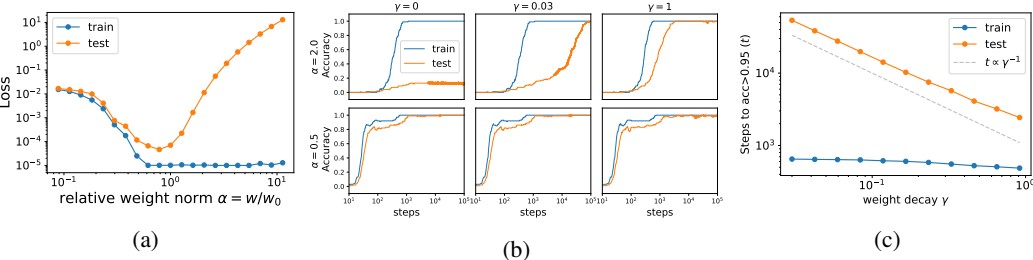

Figure 2: Teacher-student setup. $\alpha$: student initialization scale, $\gamma$: weight decay. (a) The reduced training loss and test loss have the shape of "L" and "U", respectively. (b) Top row: large initialization ($\alpha = 2.0$) can demonstrate no generalization (no reg), grokking (small reg) and fast generalization (large reg). Bottom: small initialization ($\alpha = 0.5$) always generalizes fast, regardless of weight deacy. (c) $\alpha = 2$. The steps to overfitting is independent of weight decay, while the steps to generalization scale inversely with the weight decay.

To illustrate how the LU mechanism results in grokking, we employ a toy teacher-student setup. The teacher and the student share the same architecture (a 5-100-100-5 MLP with tanh activation), but are initialized with different seeds. The student network is initialized with the standard initialization (the default one in PyTorch) but each weight is rescaled by the same factor $\alpha \equiv w/w_0$, where $w_0$ and $w$ are the weight norm of the student network before and after rescaling. The teacher network is initialized standardly, i.e., $\alpha_{\text{teacher}} = 1$. Inputs and outputs have dimensions $d_{\text{in}} = 5$ and $d_{\text{out}} = 5$, respectively. We generate $N_{\text{train}} = 100$ training and $N_{\text{test}} = 100$ test samples by first drawing inputs from the standard Gaussian distribution $N(0, \mathbf{I}_{d_{\text{in}} \times d_{\text{in}}})$, and then feed the input data to the teacher to generate output labels. The student network is trained with the Adam optimizer (learning rate $3 \times 10^{-4}$) for $10^5$ steps.

---

[2] $\gamma$ should not be too large, otherwise it will bring the weights to a trivial solution $\mathbf{w} = \mathbf{0}$.

[3] $w$ should not be too small to harm optimization.

**LU landscapes** Firstly, we compute the reduced losses by minimizing the training loss (excluding weight decay) while constraining the weight norm of the student network to be constant. We assume the converging point after training as the global minimum on the spherical surface [4], which explicitly defines the reduced losses $\tilde{l}_{\text{train}}(\alpha)$ and $\tilde{l}_{\text{test}}(\alpha)$. As shown in Figure 2a, $\tilde{l}_{\text{test}}(\alpha)$ first decreases and then increases as $\alpha$ increases, displaying a U-shape with a minimum at $\alpha \approx 1$. By contrast, $\tilde{l}_{\text{train}}(\alpha)$ decreases when $\alpha < 1$ and remains flat near zero when $\alpha \geq 1$, forming an L-shape. When weight decay $\gamma$ is present, the training landscape becomes $\tilde{l}_{\text{train}}(\alpha, \gamma) = \tilde{l}_{\text{train}}(\alpha) + \gamma \alpha^2 C^2$ where $C$ is the average parameter magnitude determined by the standard initialization.

**Training dynamics** Our problem is a regression task, but we can imitate the behavior of a classification task by manually setting a threshold $\beta = 0.01$ and defining a sample to be correctly "classified" if the prediction error is less than $\beta$. We study the dynamics of training and test accuracy. Note that this is the normal training setup where the weight norm is not constrained, although with two non-standard initializations $\alpha = 0.5$ (small) and $\alpha = 2.0$ (large), and three weight decays $\gamma = 0$ (no reg), $\gamma = 0.03$ (small reg) and $\gamma = 1$ (large reg). As shown in Figure 2b (bottom), small initialization runs always generalize fast regardless of regularization. Large initialzation runs (top) dependend on weight decay: no regularization fails to generalize, small regularization generalizes slowly (grokking), while large regularization generalizes faster.

For the large initialization $\alpha = 2.0$, we do a finer sweep of $\gamma$ in $[0.03, 1]$. We compute the number of steps and weight norm $w$ when training or test accuracy reaches 95%. As shown in Figure 2c, the time (number of steps) to reach 95% training accuracy is independent of weight decay $\gamma$, while the time to reach 95% test accuracy is inversely proportional to the weight decay, as we derived above for the LU mechanism.

# 4 OMNIGROK: GROKKING FOR MORE INTERESTING TASKS

We now analyze loss landscapes and search for grokking for several more interesting datasets, and see that the insights obtained from our toy model can transfer to these datasets. We report the main results here, with experiment details included in Appendix A.

**Image classification** We visualize loss landscapes of MNIST (Deng, 2012) to verify the LU mechanism, and study the dependence on training data size. Similar to the teacher-student case, we reduce losses and errors (one minus accuracy) to two variables (weight norm $w$ and data size $N$) by minimizing over angular directions of weights, i.e.,

$$\tilde{l}_{\text{train}}(w, N) \equiv l_{\text{train}}(\mathbf{w}^*, N), \quad \tilde{l}_{\text{test}}(w, N) \equiv l_{\text{test}}(\mathbf{w}^*, N), \quad \mathbf{w}^*(w, N) \equiv \underset{||\mathbf{w}||_2 = w}{\operatorname{argmin}} \, l_{\text{train}}(\mathbf{w}, N), \quad (3)$$

shown in Figure 3 (a)(b). The reduced loss landscape reveals three things: (1) Larger initializations lead to grokking. Point **A** in Figure 3 corresponds to the standard initialization ($\alpha = 1$), which has low training and test errors, hence no grokking. When increasing the weight norm from **A** to **B**, training error is seen to remain low while test error rises. To generalize, weight decay must be in place to bring the weight norm down, leading to grokking if weight decay is small. (2) Larger datasets lead to de-grokking. Comparing **B** and **C** in Figure 3, **C** is seen to have larger training size than **B** and lower test error. Larger data size $N$ makes the Goldilocks zone broader, reducing or eliminating grokking even for large weight initializations. (3) Critical data size can be defined. As reported in Power et al. (2022); Liu et al. (2022), we see that there exists a critical training set size below which generalization is impossible. The effective theory analysis in Liu et al. (2022) only applies to algorithmic datasets, but not to other datasets with unknown optimal representations. The loss landscape analysis presented is this work can apply to all supervised-learning tasks. As shown in Figure 3 (b), the contours of constant test error are thumb-like, and the tip of the thumb determines the minimum amount of data required for generalization.

Guided by the landscape analysis, we make two nonstandard decisions to induce grokking on MNIST: (1) we reduce the size of the training set from 60k to 1k samples (by taking a random subset) and (2) we increase the scale of the weight initialization distribution (by multiplying the initial weights, sampled with Kaiming uniform initialization, by a constant $\alpha > 1$). With these modifications to the

---

[4]This is generally not true when the loss landscape is non-convex. The aim of this assumption is to make the minimizer aligned with Eq. (1).

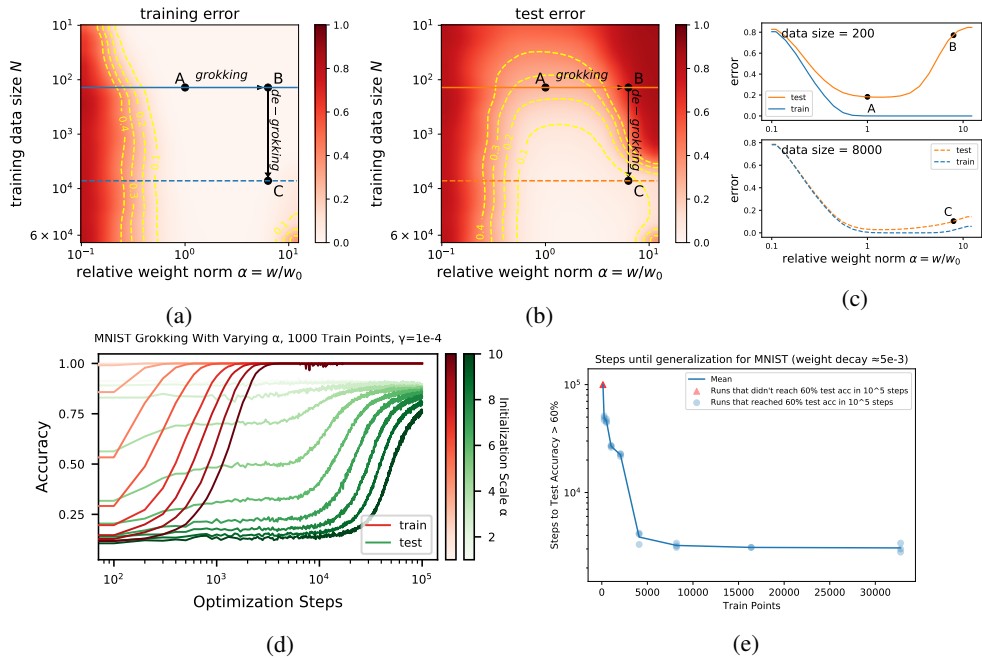

Figure 3: MNIST. (a) reduced training error, (b) reduced test error. Comparing A and B: larger weight norm makes learning grok (delay generalization). Comparing B and C: a larger training data size makes learning de-grok (speed up generalization). (c) "LU" holds truer for smaller data. (d) Accuracy curves for MNIST in the setting where we observe grokking. (e) Time to generalize as a function of training set size $N$, replicating Liu et al. (2022).

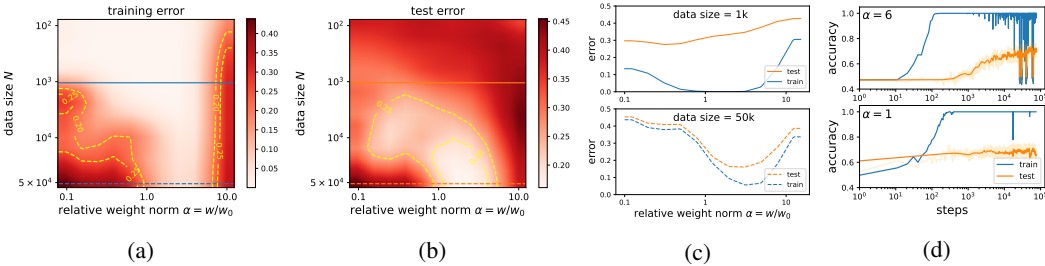

Figure 4: We use an LSTM to predict IMDb reviews. (a) training error; (b) test error; (c) reduced losses for data size 1k (top) and 50k (bottom); (d) With 1k data, a (weak) grokking signal is observed for large initializations ($\alpha = 6$), while no grokking is observed for standard initializations ($\alpha = 1$).

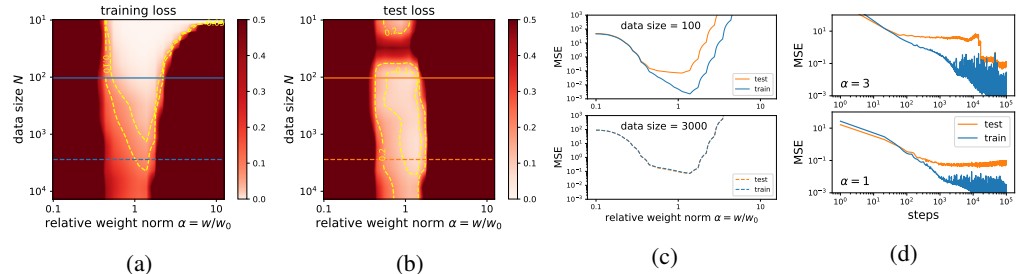

Figure 5: We use a GCNN to predict isotropic polarizability of molecules in the QM9 dataset. (a) training loss; (b) test loss; (c) reduced losses for data size 100 (top) and 3000 (bottom); (d) with 200 training samples, grokking is observed for large initialization ($\alpha = 3$).

training set size and initialization scale, we train a depth-3 width-200 MLP with ReLU activations with the AdamW optimizer using MSE loss with one-hot targets. We find that the network quickly fits the training set, and test accuracy improves much later, as shown in Figure 3d, just as in the stereotypical grokking learning first observed in algorithmic datasets. Figure 3e shows the effect of training set size on time to generalization for MNIST. We find a result similar to what Power et al. (2022) observed, namely that generalization time increases rapidly once one approaches a certain critical data set size. The conclusions still hold for the cross entropy loss (see Appendix F), although with quantitatively milder effects.

**Sentiment analysis of text** We look for grokking using LSTMs (Hochreiter and Schmidhuber, 1997) for IMDb dataset (Maas et al., 2011). Similar to Eq. (3), we reduce training and test losses to depend on only the weight norm $w$ and data size $N$. We show the reduced training and test error in Figure 4 (a)(b). For large data size, e.g., the full dataset, training and test errors have similar "U" shapes [5], so one cannot create grokking via the "LU" mechanism. For small data size, say 1k, however, the mismatch between training and test errors makes it possible to create grokking via large initializations. In Figure 4 (c), we initialize weights larger ($\alpha = 6$) with weight decay 1, overfitting is complete within $10^2$ steps, but generalization does not start until around $10^3$ steps. Note that the generalization "jump" is not as sharp as on algorithmic datasets (Power et al., 2022) or MNIST, but at least generalization is delayed here. By contrast, if we use the standard initialization ($\alpha = 1$) with no weight decay, generalization happens early on during training, and does not improve much after overfitting.

**Molecules** We search for grokking using the graph convolutional neural network (GCNN) for QM9 dataset (Ramakrishnan et al., 2014). Similar to Eq. (3), we define the reduced training/test losses, which are only dependent on weight norm $w$ and data size $N$. As shown in Figure 5(a)(b), when data size is large, training and test losses have similar "U" shapes, hence grokking is impossible via the "LU mechanism". When data size is small, training and test losses mismatch somewhere in the region $\alpha = w/w_0 > 1$, making grokking possible. Indeed, shown in Figure 5(d), there is a sharp drop in test loss around $10^4$ steps if initialization is 3 times larger than standard, while standard initialization does not lead to grokking. Note that zero weight decay is applied in both cases, implying the existence of implicit regularizations.

## 5 REPRESENTATION IS KEY TO GROKKING

In Section 4, we showed that increasing initialization scales can make grokking happen for standard ML tasks. However, this seems a bit artificial and does not explain why standard initialization leads to grokking on algorithmic datasets, but not on standard ML datasets, say MNIST. The key difference is how much the task relies on representation learning. For the MNIST dataset, the quality of representation determines whether the test accuracy is 95% or 100%; by contrast in algorithmic datasets, the quality of representation determines whether test accuracy is random guess (bad representation) or 100% (good representation). So overfitting (under a bad representation) has a more dramatic effect on algorithmic datasets, i.e., the model weights increase quickly during overfitting but test accuracy remains low. During overfitting, model weight norm is much larger than at initialization, but then drops below the initialization norm when the model generalizes, shown in Figure 9 (see Appendix C), and also observed by Nanda et al. (2023).

In the following, we will compare algorithmic datasets (Section 5.1) to MNIST (Section 5.2). We show how their loss landscapes depend on representations differently, and how the difference leads to different outcomes (grokking or not).

### 5.1 ALGORITHMIC DATASETS

**Setup** Algorithmic datasets are the task of learning a binary operation $a \circ b = c$ ($a, b, c$ are symbols) with neural networks, which aim to predict $c$ from input $(a, b)$. We take the toy addition setup in (Liu et al., 2022), where each input digit $0 \le i \le p - 1$ (output label $0 \le k \le 2(q - 1)$) is embedded as a vector $\mathbf{E}_i$ ($\mathbf{Y}_k$). A decoder MLP is employed to predict $\mathbf{Y}_k = \text{Dec}(\mathbf{E}_i + \mathbf{E}_j)$ ($k = i + j$). In the setup of grokking, both the decoder and the input representations $\mathbf{R} \equiv \{\mathbf{E}_i\}$ are trainable, with learning

---

[5]In principle, reduced training losses should be non-increasing ("L"), but optimization issues may occur for too large initializations (Schoenholz et al., 2016).

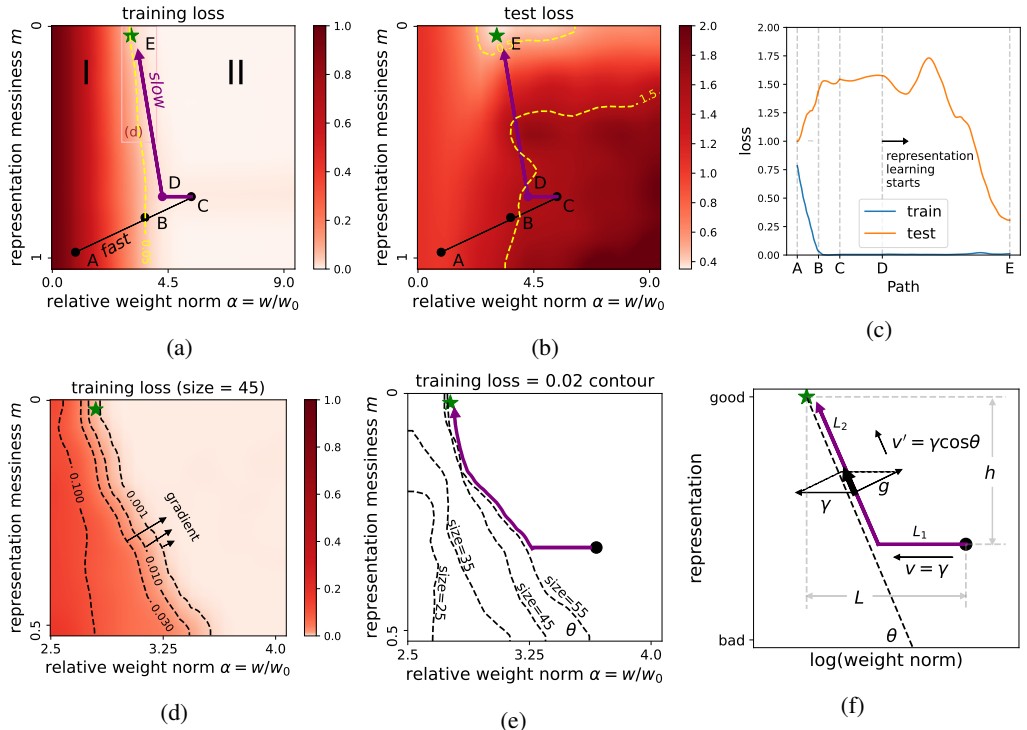

Figure 6: Loss landscapes on the 2D $(w, m)$ plane. (a) Training loss splits the plane into two regions: large loss small $w$ (fast dynamics) and small loss large $w$ (slow dynamics). (b) Test loss; the green star is the generalizing solution. (c) Losses along an illustrative path A $\rightarrow$ E, demonstrating multiple descent; (d) zoom-in of the training loss highlighting the gradients on the boundary. (e) the boundary depends on training data size; (f) a simple illustration of grokking dynamics.

rates $\eta_D$ and $\eta_R$, respectively; in the setup of landscape analysis, only decoder is trainable, as we explain below. Training and test losses depend on three factors: (i) representation $\mathbf{R}$, (ii) weight norm $w$ and (iii) weight direction $\hat{\mathbf{w}}$. As in previous sections, we can optimize $\hat{\mathbf{w}}$ by minimizing the training loss on constant weight norm spheres. We further reduce the high-dimensional representations to 1D by interpolating in a particular direction:

$$\mathbf{R} = m\mathbf{R}_{\text{random}} + (1 - m)\mathbf{R}_{\text{linear}} \tag{4}$$

where $\mathbf{R}_{\text{linear}}$ refers to the linear representation in which number $k$ is embedded to $\mathbf{E}_k = [k, 0, \cdots, 0]$, $\mathbf{R}_{\text{random}}$ is the initialized representation drawn from Gaussian distributions, i.e, $\mathbf{E}_k \sim N(\mathbf{0}, \mathbf{I})$, and $m \in [0, 1]$ is a scalar interpolating between $\mathbf{R}_{\text{linear}}$ and $\mathbf{R}_{\text{random}}$, that we term *representation messiness* because $\mathbf{R} = \mathbf{R}_{\text{linear}}$ when $m = 0$, and $\mathbf{R} = \mathbf{R}_{\text{random}}$ when $m = 1$. After these reductions, both training and test losses become functions of two variables, representation messiness $m$ and weight norm $w$:

$$\mathbf{w}^*(w, m) \equiv \underset{||\mathbf{w}||_2 = w}{\text{argmin}} \, l_{\text{train}}(\mathbf{w}, m), \quad \tilde{l}_{\text{train}}(w, m) \equiv l_{\text{train}}(\mathbf{w}^*, m), \quad \tilde{l}_{\text{test}}(w, m) \equiv l_{\text{test}}(\mathbf{w}^*, m) \tag{5}$$

Note that our definition of $\tilde{l}_{\text{train}}(w, m)$ excludes the weight decay term $\ell_{\text{reg}} = \frac{1}{2}\gamma w^2$, but we should be aware of its presence when we analyze the dynamics of $(w, m)$, which is governed by the gradient flow on $\tilde{l}_{\text{train}}(w, m)$ plus weight decay ($\eta_R/\eta_D$ are learning rates of representation/decoder):

$$\frac{dw}{dt} = -\eta_D \left( \frac{\partial \tilde{l}_{\text{train}}}{\partial w} + \gamma w \right), \quad \frac{dm}{dt} = -\eta_R \frac{\partial \tilde{l}_{\text{train}}}{\partial m}. \tag{6}$$

More experimental details are included in Appendix E.

**Landscape** We show $\tilde{l}_{\text{train}}(w, m)$ and $\tilde{l}_{\text{test}}(w, m)$ in Figures 6a and 6b, indicating the generalizing solution with a green star. Based on the reduced training loss (Figure 6a), we can divide the 2D plane

into two regions **I** and **II**, separated by a dashed yellow line (the contour of training loss = 0.05): (**I**): The darker region, with high training losses/gradients and small weight norm. (**II**): The lighter region, with low training losses/gradients and large weight norm. Comparing Figures 6a and 6b reveals that training and test loss landscapes differ, especially in region **II**. Moreover, while the training loss depends weakly on $m$, the test loss depends strongly on $m$. As we will see, the (weak) dependence of training loss on representation drives the model to the generalizing solution. However, the driving force is small because the dependence is weak, leading to grokking. We elaborate below how these particular loss landscapes lead to grokking.

**Grokking dynamics** In region **II**, the dynamics is slow (for small $\gamma$) due to nearly vanishing gradients. By contrast, the dynamics in region **I** is relatively fast. As we will explain, dynamics is also slow on the boundary of **I** and **II**, and grokking is the consequence of traversing region **II** and/or the boundary.

Let us analyze a typical path **A** to **E** shown in Figure 6(a)(b). **A** rolls "downhill" to **B** following training gradients, possibly continuing to **C** due to momentum. **C** is located in **II** where $\tilde{l}_{\text{train}} \approx 0$, so according to Eq. (6), $dm/dt \approx 0$ and $dw/dt \approx -\eta_D \gamma w$ or, equivalently, $d(\log w)/dt \approx -\eta_D \gamma$. So $(\log w, m)$ moves with a constant speed $v = \eta_D \gamma$ in the $-w$ direction from **C** to **D**, a point near the boundary. Negative gradients around the boundary point towards larger $w$ and smaller $m$, shown in Figure 6d (a zoom-in of Figure 6a). The gradients become increasingly large as the model goes deeper inside region **I**, and at some point, the gradient totally cancels out $v$ in the gradient direction, making the model start to drift along the boundary, as illustrated in Figure 6f. Then the model moves along the boundary with a new velocity $v' = v\cos\theta$ [6], until it reaches the generalizing solution **E**. The above picture is supported by empirical experiments in Appendix C and also Nanda et al. (2023). Based on the picture, we also show the ability to eliminate grokking in Appendix C.

The slow dynamics from **C** to **E** is the origin of grokking. During this period, the model first moves in the $-w$ direction with a velocity $v$ over the distance $L_1 = L - h\cot\theta$, and then moves along the boundary with a velocity $v'$ over the distance $L_2 = h/\sin\theta$. So the total time is $t = L_1/v + L_2/v' = (L + h\tan\theta)/(\eta_D \gamma)$. This formula agrees with the observation that large weight decays $\gamma$ and/or larger decoder learning rates $\eta_D$ can make generalization happen faster (Power et al., 2022; Liu et al., 2022). Besides, the path manifests intriguing multiple descent of test loss, shown in Figure 6c.

**Dependence of grokking on training data size** Another important observation in Power et al. (2022) is that grokking happens faster for larger training size. Our landscape analysis can also explain the data size dependence. In Figure 6e, we show the contours (training loss = 0.02) for different training sizes (25, 35, 45, 55). The contours of training size 45 and 55 both connect to the green star, meaning that generalization will eventually happen. However, the slopes of the contours are different, i.e., $\theta_{55} < \theta_{45}$. Since $t = (L + h\tan\theta)/(\eta_D \gamma)$ increases as $\theta$ increases, we have $t_{55} < t_{45}$, i.e, more training data leads to faster grokking. For training size 35 and 25, the contours do not connect to the green star, so generalization will not happen, no matter how long the training will be run.

## 5.2 MNIST

We now study how training and test losses depend on representation messiness in the MNIST dataset. We denote the $28 \times 28$ images as the raw representation $\mathbf{R}_{\text{raw}}$. We construct a linearly separable representation $\mathbf{R}_{\text{linear}}$ by assigning input representations proportional to their label $y_i$, for example, an image of a 2 is represented by a matrix with all elements being 2. Similar to Eq. (4), we use $m \in [0, 1]$ to interpolated between $\mathbf{R}_{\text{raw}}$ and $\mathbf{R}_{\text{linear}}$:

$$\mathbf{R} = m\mathbf{R}_{\text{raw}} + (1 - m)\mathbf{R}_{\text{linear}}, \tag{7}$$

Similarly to Eq. (5), we define and plot $\tilde{l}_{\text{train}}(w, m)$ and $\tilde{l}_{\text{test}}(w, m)$ in Figure 7, using the full dataset $N = 60000$. Comparing Figures 7a and 7b reveals two things: (1) The training and test losses behave similarly; (2) Both training and test losses depend very weakly on $m$. This implies that the raw image representation is already quite close to being optimal, so decent test accuracy can be obtained even without learning optimal representations. As a result, grokking does not occur (Figure 7c).

Comparing Figure 6 and 7, we see that the (strong) dependence of test performance on the representation is the key to grokking: the dependence on representation is strong for algorithmic datasets, so grokking happens. By contrast, the dependence is weak for MNIST, so grokking does not happen.

---

[6]For simplicity, we assume $\eta_R = \eta_D$ here, but the analysis can apply to any $(\eta_R, \eta_D)$.

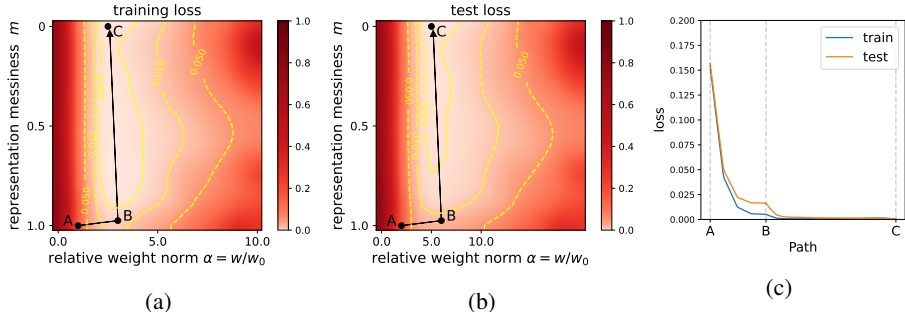

Figure 7: MNIST landscapes as functions of representation messiness $m$ and weight norm $w$: (a) training loss, and (b) test loss. Training and test losses do not have significant mismatch, and neither of them depend on representation strongly, which is in stark contrast to algorithmic datasets (Figure 6). (c) an illustrative path $A \rightarrow B \rightarrow C$ does not manifest grokking.

## 6 RELATION TO RELATED WORKS

**Grokking** was first observed for algorithmic datasets by Power et al. (2022). Several attempts have been made to understand grokking: (a) Liu et al. (2022) attributes grokking to the slow formation of good representations. (b) Shah (2021) suggests that generalizable solutions achieve lower loss than overfitting solutions, providing a training signal encouraging generalization. (c) Nanda et al. (2023) suggests grokking is a phase change due to limited data and regularization. (d) Barak et al. (2022) suggests that generalization is due not to random search, but to hidden progress of SGD to gradually amplify a Fourier gap. (e) Thilak et al. (2022) links grokking to the "Slingshot mechanism" specific to adaptive optimizers. (f) Millidge (2022) describes training as a random walk over parameters. Our conclusion supports (a)(b)(c)(d), but does not necessarily negate (e)(f).

**Double descent** is the phenomenon that performance first gets worse and then gets better as we increase the model size, data size, training epochs or regularization (Nakkiran et al., 2021; Yilmaz and Heckel, 2022; Nakkiran, 2019). The typical "U" shape of test loss in this paper does not conflict with double descent, because we are plotting the weight norm instead of the number of model parameters (Ng and Ma, 2022). However, the "U"-shape should better be considered as empirically common rather than provably universal. In fact, the interaction between properties of data and inductive biases of learning algorithms can be more complicated than double descent (Chen et al., 2021; d'Ascoli et al., 2020).

**Initialization** From the optimization perspective, initializations are usually based on the "edge of chaos" idea such that variance of features and gradients should be preserved in the forward and backward pass (Glorot and Bengio, 2010; He et al., 2015; Bahri et al., 2020; Yang and Schoenholz, 2017; Jing et al., 2017), or based on analyzing Jacobians and/or Hessians (Skorski et al., 2020). From the generalization perspective, it was shown that large initializations overfit data easily but result in poor generalization (Xu et al., 2019; Zhang et al., 2020), which agrees with our LU mechanism.

**Weight decay regularization** is a standard trick in machine learning and has various effects on optimization and generalization (Zhang et al., 2018; Van Laarhoven, 2017). In particular, Lewkowycz and Gur-Ari (2020) observes that it takes $t \propto 1/\lambda$ training steps to reach maximum test performance. This is strikingly similar to the grokking time $t \propto 1/\lambda$ we derived from the LU mechanism.

## 7 CONCLUSIONS

This study elucidates the grokking phenomenon from the perspective of loss landscapes. Our conclusions are: (i) grokking originates from the mismatch between training and test losses at high model weight norm ("LU" mechanism). (ii) grokking can happen in various models for a wide range of datasets, although the grokking signature is usually most dramatic for algorithmic datasets. (iii) The severity of grokking depends on how much the task relies on learning representations. This work not only reveals the mechanism of grokking, but also shows that reduced landscape analysis is a useful tool for characterizing data-model interaction and representation learning.

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

# Appendix

## A  EXPERIMENT DETAILS

**Sentiment analysis of text** IMDb (Maas et al., 2011) includes 50k movie reviews to be classified as being positive or negative. To pre-process the data, we extract the 1000 most frequent words and tokenize each review into an array of token indices. Less frequent words are ignored, and each review array is padded to length 500. We adopt the LSTM model (Hochreiter and Schmidhuber, 1997) to perform the classification, with two layers, embedding dimension 64, and hidden dimension 128. We use the Adam optimizer (Kingma and Ba, 2014) with learning rate 0.001 to minimize the binary cross entropy loss. We hold back 25% of the dataset for testing.

**Molecules** QM9 is a database for small molecules and their properties. We use a graph convolutional neural network (GCNN) to predict the isotropic polarizability. The GCNN contains 2 convolutional layers with ReLU activation, followed by a linear layer. We use the Adam optimizer with learning rate 0.001 to minimize the MSE loss. We split the dataset into 50/50 train/test.

**MNIST** We train width-200 depth-3 ReLU MLPs on the MNIST dataset with MSE loss. We use the AdamW optimizer with a learning rate of 0.001 and a batch size of 200.

## B  REDUCED LOSS FOR MODULAR ADDITION WITH TRANSFORMERS

In Figure 8 we show reduced loss landscape plots for transformers trained on modular addition. We use the setup of Nanda et al. (2023) and train a 1-layer transformer on modular addition ($p = 113$) with $d_{model} = 128$, 4 attention heads, and $d_{mlp} = 512$ with ReLU activations. We train with a learning rate of 0.001 while constraining model weight norm, for a variety of $\alpha$ and a variety of train set fractions. The LU shape holds for $\alpha \in [0.1, 4]$ (some optimization issue may be responsible for the rise in train loss for $\alpha > 4$). We see the critical train set size is approximately 0.25, in line with earlier studies on grokking.

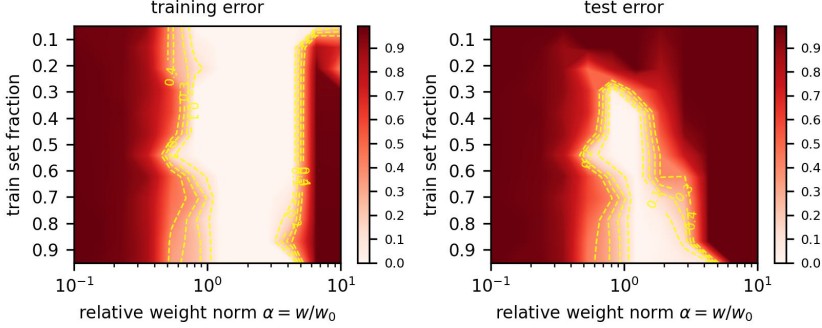

Figure 8: Reduced loss landscapes for transformers trained on modular addition, the original setting where grokking was observed.

## C  WEIGHT NORM EVOLUTION OVER TIME ON ALGORITHMIC TASKS

**Evolution of weight norm** As mentioned in Section 5, the dynamics of model weight norm over the course of training, on algorithmic tasks, support the LU mechanism picture of grokking. Figure 9a, shows how model norm changes over time and we see that there is an initial increase in weight norm, which peaks during overfitting, but then drops during the period of generalization to be lower than the initialization norm. For this experiment, we again used the setup of (Nanda et al., 2023). We train with AdamW with a learning rate of 0.001 and weight decay $\gamma = 1$.

**Constraining a small weight norm eliminates grokking** As shown in Figure 9b, reducing the initialization scale ($\alpha = 0.8$) and constraining optimization to hold model weight norm constant

over training brings train accuracy and test accuracy learning curves together, almost eliminating grokking. We would like to investigate in future works whether this training trick can be helpful for more standard machine learning tasks.

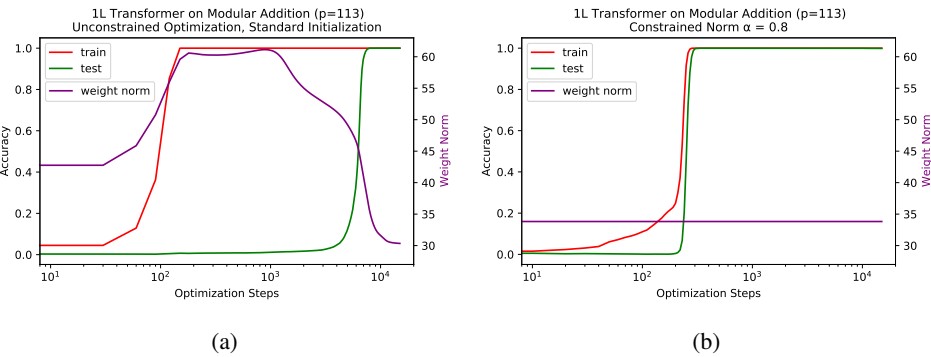

Figure 9: Training 1L transformer on modular addition ($p = 113$). (a) Weight norm, train accuracy, and test accuracy over time, initialized and trained normally. Weight norm first increases, and is highest during the period of overfitting, but then drops to become lower than initial weight norm when the model generalizes. (b) Constrained optimization at constant weight norm ($\alpha = 0.8$) largely eliminates grokking, with test and train accuracy improving almost concurrently.

## D    TIME TO GENERALIZE VERSUS WEIGHT DECAY

In our discussion of the "LU mechanism" as an explanation for grokking in Section 2, we predicted that the training time required for a model to generalize should be $t \propto \gamma^{-1}$ where $\gamma$ is the weight decay. To test this, we perform a grid search over weight decays $\gamma$ and plot the number of training steps required for models to reach a specified level of test accuracy in Figure 10a-10b. We also show full training curves for these runs in Figure 10c-10d. We perform experiments in two setups:

(a) **Transformer on modular addition**: We use the replication of grokking from Nanda et al. (2023) and train a 1-layer transformer on modular addition ($p = 113$ and a train set fraction of 0.3) where $d_{\text{model}} = 128$, with 4 attention heads, $d_{\text{mlp}} = 512$, ReLU activations, and an AdamW learning rate of 0.001. From Figure 10a, we find that $t \propto \gamma^{-1}$ holds across roughly two orders of magnitude of $t$ and $\gamma$. There is some seed dependence on the generalization time (some seeds consistently require longer to generalize), but for each seed (corresponding to a particular model initialization) the relation $t \propto \gamma^{-1}$ appears to fit the data well.

(b) **ReLU MLP on MNIST**: We train ReLU MLPs on MNIST as described in Appendix A. We use an $\alpha = 9.0$ and train on a reduced training set of 1000 samples to delay generalization / induce grokking. From Figure 10b, we find that for $\gamma$ roughly between 0.1 and 1.0 the relation $t \propto \gamma^{-1}$ holds. Very high values of weight decay seem to mess with optimization. On the other hand, with very low weight decay the model generalizes faster than naively expected, perhaps due to implicit regularization.

## E    SECTION 5.1 SETUP

**Architecture** Similar to Liu et al. (2022), the decoder architecture is an MLP with hard coded addition. Each input symbol $i$ is encoded to a scalar $E_i$. Each output symbol $k$ is represented by a 30D random vector $\hat{\mathbf{Y}}_k$. We consider addition with base $p$, so input $0 \leq i, j \leq p - 1$ and output $0 \leq k = i + j \leq 2(p - 1)$. We denote *representation* as $\mathbf{R} = \{E_0, E_1 \cdots, E_{p-1}\}$. The MLP has two hidden layers, with neurons 1-200-200-30 in each layer and ReLU activations. Given a training sample $(E_i, E_j) \to \mathbf{Y}_k$ where $i + j = k$, the prediction of the MLP decoder is

$$\mathbf{Y}_k = \text{Dec}_{\mathbf{w}}(E_i + E_j), \tag{8}$$

and the loss function being the mean squared error (MSE) between $\mathbf{Y}_k$ and $\hat{\mathbf{Y}}_k$, and $\mathbf{w}$ being the decoder weight. Although the common setup of grokking is to make both the representation $\mathbf{R}$ and

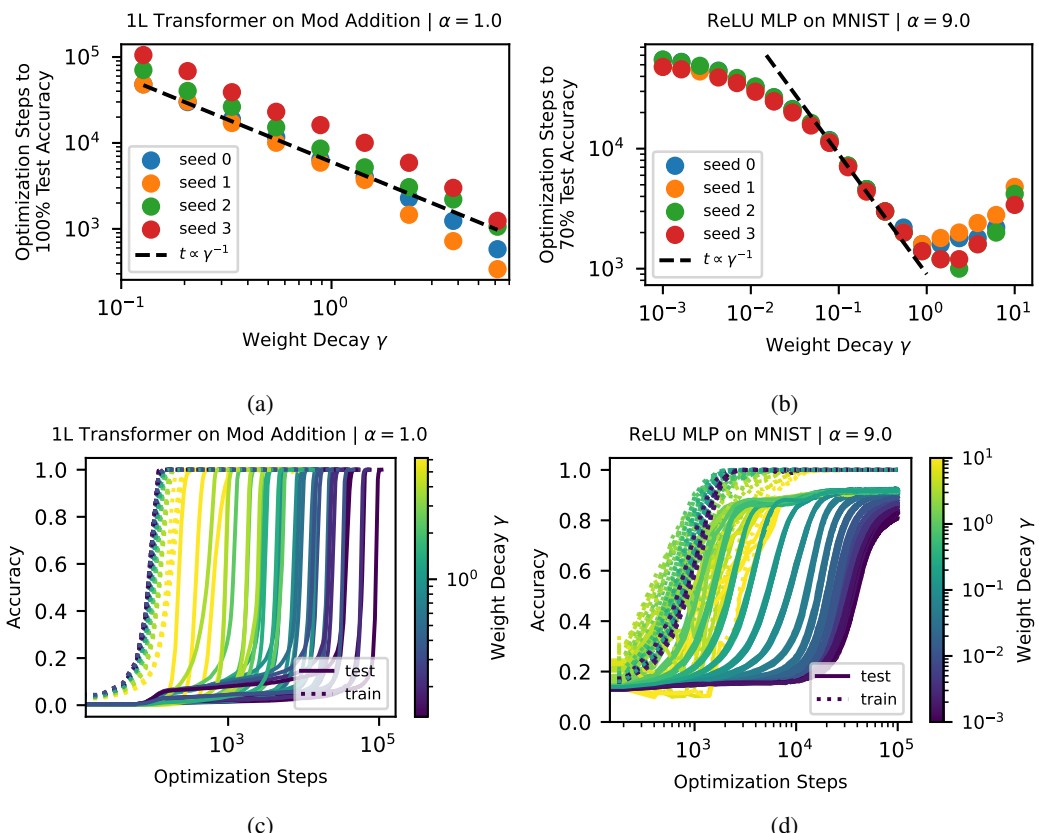

Figure 10: Time to generalize as a function of weight decay: we investigate to what extent the relation $t \propto \gamma^{-1}$ holds, where $t$ is number of training steps needed for the model to generalize and $\gamma$ is the AdamW weight decay. When a lower weight decay is used, models spend longer in the period of overfitting before eventually generalizing. We show the generalization time $t$ as a function of $\gamma$ in (a)-(b) and full training curves for these runs in (c)-(d).

| Trainability | decoder weight $\mathbf{w}$ | | Representation $\mathbf{R}$ | |
|---|---|---|---|---|
| | norm $w = \|\|\mathbf{w}\|\|_2$ | direction $\hat{\mathbf{w}} = \mathbf{w}/w$ | messiness $m$ | Other |
| Landscape analysis | No, $w$ | Yes | No, $m$ | No, 0 |
| Reduced trajectory | Yes | No, $\hat{\mathbf{w}}^*(w, m)$ | Yes | No, 0 |
| Full trajectory | Yes | Yes | Yes | Yes |

Table 1: Threes setups used in this paper, with different set of parameters trainable.

the decoder $\mathbf{w}$ trainable, we will freeze part of them for easier analysis. This is where it could be a bit confusing, so we explicitly distinguish three setups: *landscape analysis*, *reduced trajectory analysis* and *full trajectory analysis*. Each setup have different subset of trainable parameters, as shown in Table 1.

**Landscape analysis** Both the representation $\mathbf{R}$ and weight norm $w$ are fixed. Only the weight direction $\hat{\mathbf{w}}$ is trainable. The representation $\mathbf{R}$ is fixed according to Eq. (4), which is dependent on $m$, the representation messiness. The decoder has fixed weight norm $w$, but the weight direction $\hat{\mathbf{w}}$ is trainable. For each fixed $(w, m)$, we minimize training loss over $\hat{\mathbf{w}}$ to get

$$\hat{\mathbf{w}}^*(w, m) = \underset{\hat{\mathbf{w}}}{\mathrm{argmin}} \, \ell_{\mathrm{train}}(w, m, \hat{\mathbf{w}}), \tag{9}$$

and define reduced training and test loss, as in Eq. (5). The minimization is implemented by the Adam optimizer with learning rate $10^{-3}$ for $10^4$ steps. Although $(w, m)$ are not trainable, we repeat the above minimization independently for different $(w, m)$. In Figure 6 (a)(b)(d), the background heatmaps belong to landscape analysis.

**Reduced trajectory analysis** is a "thought experiment" based on landscape analysis. Since full trajectory analysis can be intractable due to too high dimensions, we try to reduce the trajectory anaysis to 2D, by making two assumptions about the real dynamics: (1) *Scale separation*: the dynamics of $\hat{\mathbf{w}}$ is much faster than the dynamics along $w$ and along $m$, such that $\hat{\mathbf{w}}(t) = \hat{\mathbf{w}}^*(w(t), m(t))$ is valid at every moment during training. (2) *Representation evolution is linear*, i.e., interpolating between initial random Gaussian and final linear representation. With these two assumptions, the training dynamics is effectively reduced to 2D, depending only on $(w, m)$, obeying Eq. (6). In Figure 6 (a)(b)(c), the path A $\rightarrow$ E belongs to reduced trajectory analysis.

Admittedly the reduced trajectory may deviate from the full trajectory since the assumptions may not be met, but it can shed light on the full trajectory: the weight norm first increases and then increases, and the decrease of weight norm is highly correlated with generalization (please see Appendix C and Figure 9.

## F  MNIST EXPERIMENTS WITH CROSS ENTROPY LOSS

To respond to a reviewer's concern that our use of the MSE loss is the "secret" to get grokking on MNIST (Figure 3), we reran our experiments with the cross entropy (CE) loss. The results are qualitatively similar, with some quantitative differences.

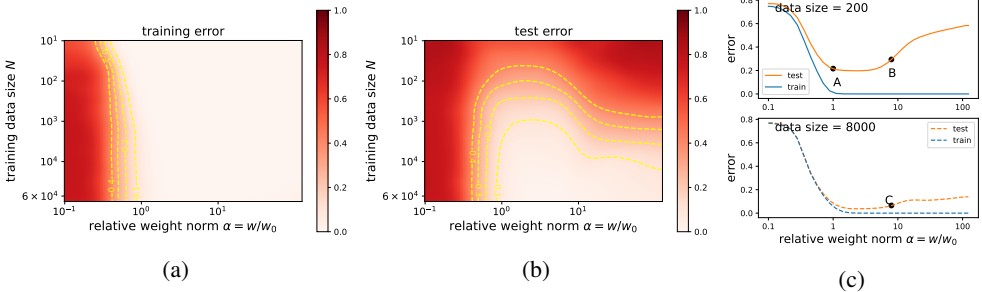

(a)                                  (b)                                  (c)

Figure 11: MNIST with the cross entropy loss (as opposed to the MSE loss used in Figure 3). (a) reduced training error, (b) reduced test error. (c) "LU" still holds for the cross entropy loss, but the effect is milder than the MSE loss. In particular, the "Goldilocks zone" (the weight range where generalization happens) is broader.

**Landscape analysis**

Comparing Figure 3 (MSE) and Figure 11 (CE), we notice the they are qualitatively similar: (1) for small datasets, the reduced training error and test error resemble an "L" and "U" against the weight norm, respectively; (2) for large datasets, the "U" becomes more like "L", i.e., the mismatch between the reduced training and test error is small. However, a quantitative difference exist: CE produces a broader "Goldilocks zone" (the weight range where generalization happens) than MSE. This implies that to induce grokking with CE, we need to increase the weight norm to a larger value (say $\alpha = 100$).

**Training dynamics**

We are able to observe delayed generalization during trianing on MNIST with cross entropy loss, but doing so requires a higher $\alpha$ than was necessary when using MSE loss, as predicted by the reduced loss landscapes in Figure 11. Figure 12 shows training trajectories from a 3-layer ReLU MLP on MNIST trained with cross entropy loss with $\alpha = 100$ and $D = 200$. We see that test accuracy rises to 30-40% early in training, then plateaus for an extended period, before increasing to $\approx 75\%$ while train accuracy remains at 100%. While the dynamics are not as clean as with MSE loss, since test accuracy first plateaus at better-than-random accuracy, we think it is still fair to classify these dynamics as "grokking" due to the improvement in generalization late in training after a plateau.

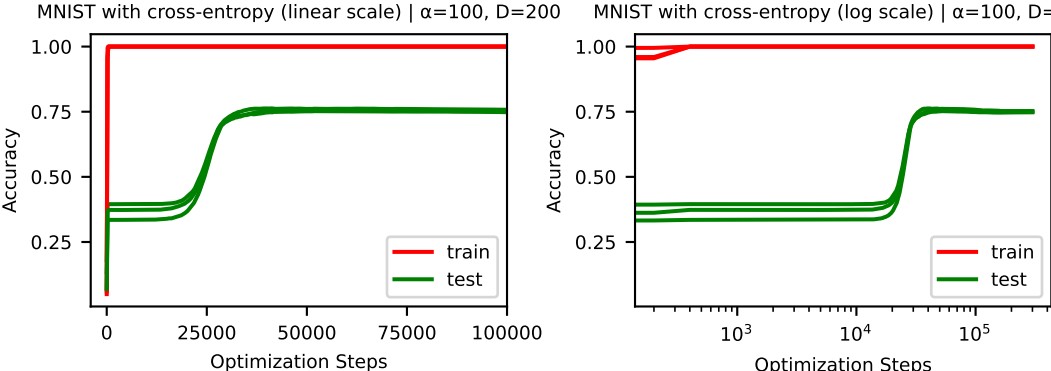

Figure 12: Training curves using cross entropy loss on MNIST. We are still able to observe delayed generalization on MNIST using cross entropy loss, though test accuracy first plateaus at higher than random-guess accuracy.

