# OpenReview forum: "Omnigrok: Grokking Beyond Algorithmic Data"
_ICLR.cc/2023/Conference — ICLR 2023 notable top 25%_

### Official Review · Reviewer_wgE4 · 2022-10-18

**Confidence:** 4
**Correctness:** 3
**Technical Novelty And Significance:** 2
**Empirical Novelty And Significance:** 3
**Recommendation:** 8

**Clarity, Quality, Novelty And Reproducibility:**

The paper is extremely clear and seems to mostly be very reproducible, though I would like to see the footnote about pytorch modified to include details of the default initialization. The paper is of very high quality in general.

The novelty is middling, with some new ideas and some old ideas presented as new ideas (LU mechanism).

**Strength And Weaknesses:**

Strengths:
- This paper is well written and presented, a pleasure to read.
- The experiments make the claims clear.
- They successfully induce delayed generalization in natural datasets, suggesting that their claims might apply in realistic settings. These results happen outside of the teacher students set in, so they are more convincing than the algorithmic experiments.
- The observation that larger dataset sizes expands the goldilocks zone is a very tidy way of thinking about generalization.
- I found myself fairly convinced by the experiments that initialization magnitude is at least one factor that can induce grokking.
- It’s good that they discuss the impact of weight decay repeatedly, as it is clearly an important factor.

Weaknesses:
- I do not generally consider it necessary to have a theoretical component of every empirical study, and many empirical studies make strong and intriguing claims that layout groundwork for a principled understanding. However, I think that the particular claims made in this paper would benefit enormously from a theoretical analysis. My issue is that they make out the initialization norm to be of particular significance, when other hyperparameters might (in fact, empirically do) have an equivalent “goldilocks zone”. Additional experiments about other hyperparameters might help, but I think that the insights that we can glean from purely empirical study of this phenomenon are limited.
- I feel that the connection between overfitting and grokking is fairly obvious, as their own definition of grokking makes clear. Given the way that they implicitly define grokking, I don’t believe that they can describe the LU mechanism as a “cause” rather than as a rephrasing of the same definition.
- These settings used to illicit grokking are somewhat artificial: the initial norm, the constrained search space, the weight decay requirement. There is no satisfactory explanation given for why weight decay is so crucial; this is where a theoretical analysis might have been helpful.
- Teacher student setups can fundamentally change training dynamics, so I’m not confident that the results from those experiments generalize to other settings. If the claims made from these experiments do not apply outside of a teacher student setting, I’m not sure I buy them. Especially because the particular representations learned by the teacher could easily require its own particular magnitude that is not necessary for all representations.
- It is not clear to me what differentiates the LU mechanism from the existing concept of overfitting. Although they described their setup as different from ‘related phenomena’  (Schoenholz et al., 2016; Yang and Schoenholz, 2017; Nakkiran et al., 2021), it’s not clear to me how these citations are not directly describing the LU mechanism.
- The effect that this paper attributes to the importance of representation learning (section 5) might be more precisely described as the effect of the distribution shift at inference time.
- “We treat the converging point after training as the global minimum on the spherical surface” — why? Is this actually a convex a surface somehow?
- It appears to me that the interpolations are between the final learned model and a random gaussian. It seems to me these experiments and plots would be more informative if the random model was the initialization weights, so that we could see a planar view of the path from initialization and not just any random direction. I personally would find that more convincing it as an explanation of the training process, compared with a random direction.
- On importance of representation learning: it seems to me that this might best be described more precisely as the significance of distribution shift between train and test time. Otherwise, I would expect you to formalize what it would mean to have representation learning be important.
- The discussion section: “(ii) Pre-training avoids learning representations from scratch, hence helps reduce possible grokking.” But why wouldn’t we see grokking in the pretraining stage itself?
- Is it definitely the case that grokking is not observed in language models if you use larger initializations? Otherwise, you should probably remove the discussion section on language models entirely or present evidence that grokking does not happen at any stage. I don't think there's anything backed up in this section, and it doesn't really relate to the rest of the results.

Minor/references:
- You refer to default initialization settings in pytorch, but I think you need to give details about what those defaults are, as they might change between versions.
- Footnote 2: I’m not sure this footnote should be there. It is well known that increased capacity allows tighter interpolation of the training set, but justifying this statement by presenting an artificial scenario in which a larger model is equivalent to the smaller model is not a reflection of real practices, where we would not initialize zero-vectors when expanding an architecture.
- There is an existing literature on distinctions between multi epoch vs ideal infinite data sampling in practice, which might be worth digging into and adding to your citations, eg Nakkiran (2019) https://arxiv.org/abs/1912.07242
- “We run experiments with two initializations α = 0.5 (small) and α = 2.0 (large)” — Above, it said that alpha was the constant weight norm throughout training, but here it says that it is only the initialization. I can see now that this is supposed to just be the weight constraint, but it’s a little confusing in the wording.
- A number of these results, especially in the section on LU, seem to boil down to demonstrating that overfitting does exist. I’m not sure that this is necessary to make such a focus.
- Formula 4 should have a citation to existing work on sharpness or linear interpolation.
- I’m a little confused about the meaning of v′ = vcosθ, given that θ was previously defined as a thresholding value for the regression model, and I don’t see how that definition connects to the use the variable in section 5.1.
- This paper uses natbib. When you make an inline citation, e.g., “Foo et al. (2009) claim…” you should use \citet{} instead of \cite{} or  \citep{} . An example of this issue is the entire first paragraph of section 6.

**Summary Of The Paper:**

This paper proposes an explanation for when grokking happens---that it occurs when the initialization norm is too large so the model takes longer to get to the correct norm magnitude—and illustrates their thesis with results on synthetic and natural data. Defining grokking as a delay in generalization until after training set interpolation, they propose that grokking is caused by a mismatch between training and test loss curves, ie, overfitting, and described this pattern as the “LU mechanism”. They elicit grokking phenomena on natural data (IMDB and MNIST), at least if we define grokking as a delayed generalization curve.

**Summary Of The Review:**

I found many of the analyses here intriguing, and ultimately I would very much like to see this paper published. I was particularly impressed by their ability to induce grokking in a realistic setting. However, they seem to be making the claim that grokking is in general caused by initializing with a large norm. The experiments do not fully support that this is the case in the main setting in which grokking has been observed, which is the algorithmic setting. I would welcome more experiments around a wider set of hyperparameters, to demonstrate that delayed generalization does not result from other settings. Without these experiments or an added theoretical analysis, I don't feel confident recommending acceptance, but I'm ready to change my mind in response to a small number of additional experiments: evidence that the effects observed are particular to the norm magnitude and illustrations of the messiness plots but with the initialization weights instead of random weights, so that we can see something closer to the actual trajectory of training. Furthermore, I think that the paper should be somewhat modified in its definitions and reasoning, as the LU mechanism appears to fall almost trivially from their definition of grokking, rather than being an attributed cause.

**After discussion:**

I'm raising my score from 6 to 8 to reflect expanded experiments that lend some additional credence to the link between behavior in the algorithmic and natural settings, and the removal and finetuning of some unsupported speculation and imprecise connections.

---

> ### Author Response · Authors · 2022-11-14
> **Response to wgE4 (Part I)**
>
> Dear reviewer wgE4, thank you for the detailed review and so many constructive questions/suggestions! Here is our response to your questions and concerns.
>
> Q0: Suggested experiments:
> * (1) evidence that the effects observed are particular to the norm magnitude;
> * (2) illustrations of the messiness plots but with the initialization weights instead of random weights;
> * (3) generalize teacher-student setup to other tasks.
>
> A0: Thank you for the suggestions!
> * (1a) In Appendix B Figure 8a, we trace the evolution of weight norm in the realistic grokking setup (transformer + modular addition), finding that: the weight norm increases and then decreases, and the decrease of weight norm is highly correlated with generalization. This agrees with our analysis of the toy setup in Section 5.
> * (1b) In Appendix B Figure 8b, we show that grokking can be eliminated by initializing and constraining the model weight to a smaller norm. This observation strengthens the  support for the LU mechanism.
> * (2) We are sorry for the confusion. We indeed use initialization weights instead of random weights. Please see Q7 for more details.
> * (3) We find that the relation $t\propto \gamma^{-1}$ (generalization steps $t$, weight decay $\gamma$) not only applies to the teacher student setup, but also generalizes to algorithmic datasets and MNIST. Please see Q3 and Appendix C for more details.
>
>
> Q1: LU mechanism is more like rephrasing of grokking, rather than a cause.
>
> A1: Thanks, this is a reasonable concern.
> * (1) We have toned down our argument in the introduction to “Grokking can result from a mismatch between training and test loss against model weight norm.” which is previously “Grokking is caused by …”.
> * (2) We agree that  “grokking” = “LU curves along *some direction*” is very intuitive. Our contribution is to point out that weight norm (rather than angular directions) is *the direction*, at least one of the directions.
> * (3) The new experiment on eliminating grokking (Appendix B Figure 8a) further strengthens the argument that grokking and model weight norm are tied.
>
> Q2: I think that the particular claims made in this paper would benefit enormously from a theoretical analysis. There is no satisfactory explanation given for why weight decay is so crucial; this is where a theoretical analysis might have been helpful.
>
> A2: We agree that lack of a rigorous theory is one limitation of our work, which we would love to investigate in the future.
>
> Q3: Teacher student setups can fundamentally change training dynamics, so I’m not confident that the results from those experiments generalize to other settings.
>
> A3: We did extra experiments finding the basic result in the teacher-student setup can transfer to realistic tasks. In particular, the relation “generalization time is inversely proportional to weight decay” can generalize to MNIST and algorithmic datasets, which we show in Appendix B.
>
> Q4: It’s not clear to me how these citations are not directly describing the LU mechanism: Schoenholz et al., 2016; Yang and Schoenholz, 2017; Nakkiran et al., 2021.
>
> A4: [Schoenholz et al., 2016] and [Yang and Schoenholz, 2017] discussed how initializations would affect training dynamics (the “L” part); [Nakkiran et al., 2021] discussed double descent of test curves (similar to the “U” part, but not exactly the same). Their difference is resolved by [Ng and Ma, 2022] showing that a  “U” curve can be recovered from a double descent curve by plotting weight norm as x axis instead of the number of parameters. We agree that the “LU mechanism” is not very novel in the literature and is a mixture of many ideas. Our contribution is to explicitly describe this picture (minor), and more importantly, use the picture to understand grokking. We hope this intuitive picture will be a fruitful way to think about neural network generalization in general.
>
> Q5: The effect that this paper attributes to the importance of representation learning (section 5) might be more precisely described as the effect of the distribution shift at inference time. Otherwise, I would expect you to formalize what it would mean to have representation learning be important.
>
> A5: We agree that “distribution shift” is a more common way to think about generalization at inference time. The reason why we term it “representation learning” is because we find it is particularly useful and intuitive for algorithmic datasets. We follow the argument in [Liu et al. 2022] (https://arxiv.org/pdf/2205.10343.pdf) that “representation is key to grokking” in which their figure 1 shows the emergence of a beautiful ring structure after generalization. The correlation between generalization and representation is what we mean by “representation learning is important”.

---

> > ### Author Response · Authors · 2022-11-14
> > **Response to wgE4 (Part II)**
> >
> > Q6: “We treat the converging point after training as the global minimum on the spherical surface” — why? Is this actually a convex surface somehow?
> >
> > A6: We apologize for this misleading argument. We have changed “treat” to “assume”, and added a footnote saying that “This is generally not true when the loss landscape is non-convex. The aim of this assumption is to make the minimizer aligned with Eq. (1).”
> >
> > Q7: It appears to me that the interpolations are between the final learned model and a random gaussian. It seems to me these experiments and plots would be more informative if the random model was the initialization weights.
> >
> > A7: We feel sorry for the confusion. We are actually using the initialization weights, although these initialization weights are initialized as random Gaussian. We refer to them as the same thing. Below Eq. (4), we emphasize that “R is the initialized representation drawn from Gaussian distributions”.
> >
> > Q8: The discussion section: “(ii) Pre-training avoids learning representations from scratch, hence helps reduce possible grokking.” But why wouldn’t we see grokking in the pretraining stage itself?
> >
> > A8: That’s a good point! We conjecture that for language models there might be many small grokkings (for subtasks) going on, but they might average out, giving rise to a smooth test curve. This is highly speculative though, so have removed the whole discussion, as suggested by the reviewer.
> >
> > Q9: Is it definitely the case that grokking is not observed in language models if you use larger initializations? Otherwise, you should probably remove the discussion section on language models entirely…
> >
> > A9: We are not sure if grokking can be observed on large language models. We agree to remove this discussion since it is not very well supported.
> >
> > Q10: You refer to default initialization settings in pytorch, but I think you need to give details about what those defaults are, as they might change between versions.
> >
> > A10: Thanks for the suggestion! We have added a footnote on page 2 that “For linear layers, each weight $w\sim U[-\sigma,\sigma]$ and bias $b\sim U[-\sigma,\sigma]$ where $\sigma = 1/\sqrt{\rm fan\\_in}$, and $U[a,b]$ denotes uniform distribution on $[a,b]$.”.
> >
> > Q11: Footnote 2: I’m not sure this footnote should be there. It is well known that increased capacity allows tighter interpolation of the training set, but justifying this statement by presenting an artificial scenario in which a larger model is equivalent to the smaller model is not a reflection of real practices, where we would not initialize zero-vectors when expanding an architecture.
> >
> > A11: Thanks! We agree that this sounds a bit artificial and have removed this footnote.
> >
> > Q12: There is an existing literature on distinctions between multi epoch vs ideal infinite data sampling in practice, which might be worth digging into and adding to your citations, eg Nakkiran (2019) https://arxiv.org/abs/1912.07242.
> >
> > A12: Thanks for pointing us to this great note! We have added it in the double descent paragraph in related works.
> >
> > Q13: “We run experiments with two initializations α = 0.5 (small) and α = 2.0 (large)” — Above, it said that alpha was the constant weight norm throughout training, but here it says that it is only the initialization. I can see now that this is supposed to just be the weight constraint, but it’s a little confusing in the wording.
> >
> > A13: We apologize for the confusion. When we reduce the loss landscapes, the weight norm is constrained. However, when we induce grokking, the weight norm is not constrained although with non-standard (0.5 or 2) initializations. We have added this sentence to clarify: “Note that this is the normal training setup where the weight norm is not constrained, although with two non-standard initializations $\alpha=0.5$ (small) and $\alpha=2.0$ (large), …”.
> >
> > Q14: A number of these results, especially in the section on LU, seem to boil down to demonstrating that overfitting does exist. I’m not sure that this is necessary to make such a focus.
> >
> > A14: We agree that LU may not be novel, and may have alternative explanations. However, we believe that LU is more than just demonstrating the existence of overfitting. Firstly, LU sheds light on how far away are overfitting solutions to generalizable solutions, and why weight decay helps. Secondly, the expansion of the goldilocks zone for larger datasets is a visual and intuitive way to quantify the dependence on data size.
> >
> > Q15: Formula 4 should have a citation to existing work on sharpness or linear interpolation.
> >
> > A15: Sorry we don’t quite see the connection. Could you please point us to some reference, if possible?

---

> > > ### Author Response · Authors · 2022-11-14
> > > **Response to wgE4 (Part III)**
> > >
> > > Q16: I’m a little confused about the meaning of v′ = vcosθ, given that θ was previously defined as a thresholding value for the regression model, and I don’t see how that definition connects to the use the variable in section 5.1.
> > >
> > > A16: Sorry for the confusion.  $\theta$ here refers to the angle between the training isocurve and the x axis, shown at the bottom of Figure 6 (f). To avoid reuse of the symbol, we have changed the thresholding value to $\beta$.
> > >
> > > Q17: This paper uses natbib. When you make an inline citation, e.g., “Foo et al. (2009) claim…” you should use \citet{} instead of \cite{} or \citep{} . An example of this issue is the entire first paragraph of section 6.
> > >
> > > A17: Thanks! We have corrected as suggested throughout the paper.

---

> > > > ### Comment · Reviewer_wgE4 · 2022-11-16
> > > > **adjusting score**
> > > >
> > > > I think what you have now is a lot more precise and less speculative than the original version. Thanks for making those modifications; I think what's left is sufficiently valuable to the community that you don't need any "spice" from unsupported speculation. I recommend contrasting the respective definitions of LU and grokking for clarity, as I mention in my other response, but I'm raising my score for the updated version.

---

> > ### Comment · Reviewer_wgE4 · 2022-11-16
> > **response**
> >
> > > (1) We have toned down our argument in the introduction to “Grokking can result from a mismatch between training and test loss against model weight norm.” which is previously “Grokking is caused by …”.
> >
> > I don't feel that this addresses my concern. Grokking does not result from the LU mechanism; rather, it seems that you have defined it in terms of the LU mechanism. Am I misunderstanding? Is there a situation where the LU mechanism appears but you wouldn't call it grokking by definition? Perhaps what you need to do is redefine the LU mechanism to apply only along a constant weight norm, which wasn't my understanding from the definition offered in the paper. Then the connection between grokking and the mechanism would no longer be tautological. (This may be what you originally intended to be understood.) It could be more clear if you explicitly defined what you meant by both grokking and the LU mechanism so the difference between them is clear. Otherwise, I think you should rephrase this to remove any mention of cause or result, and just give an explicit definition of what you consider to be grokking, which you can then link to the weight norm.
> >
> > (1b) This does strengthen the support for your claims, thanks. Appendix B also addresses some of my concerns about the link between behavior in the algorithmic and natural datasets.
> >
> > > We follow the argument in [Liu et al. 2022] (https://arxiv.org/pdf/2205.10343.pdf) that “representation is key to grokking”
> >
> > Without a more precise definition of what it means for representation to be more or less important for a particular task, I don't feel that you have any evidence for this claim. You would need to demonstrate not only that representation is key to grokking (which I accept) but that in some relevant sense it is *not* key to the other settings you are contrasting with grokking.
> >
> > It's entirely reasonable for you to publish this without a theoretical analysis, I understand that's out of scope. Mainly I wanted to have some kind of justification as to why the norm is "the" explanation. I'm still not entirely convinced on that front, but the paper is really improved by the new Appendix B, and would be further improved by adjusting some of the remaining informal / less precise / less evidenced arguments (representation learning being key in some settings and not others, the idea that the LU mechanism is a cause rather than definition of grokking).

---

### Official Review · Reviewer_ENpU · 2022-10-21

**Confidence:** 4
**Correctness:** 4
**Technical Novelty And Significance:** 3
**Empirical Novelty And Significance:** 3
**Recommendation:** 8

**Clarity, Quality, Novelty And Reproducibility:**

The paper is mostly clear and well-written. To me, the reduced 1D loss is the strongest contribution of the paper, since it is a general tool for loss landscape analysis, and we do not have enough of those. The link to representation learning is interesting, although, just like with grokking in general, I am left with a “so what?” question. Nonetheless, this is most definitely an original, interesting work.

**Strength And Weaknesses:**

Strengths:

1) The loss landscape perspective is refreshing and very tangible, I think this is the right way to think about the problem of grokking.
2) Reduced 1D loss landscapes are a powerful visualisation tool, and may find further uses in the NN analysis and understanding.
3) Empirical results are convincing.

Weaknesses:
1) The authors refer to “standard initialisation”, and then explain that it stands for default initialisation in PyTorch. While this can be easily looked up at the time of writing, I am not in favour of such loose definitions. What if the next version of PyTorch picks a different default? Surely that should not lead to non-reproducible results. As such, I would like to request the authors to rather define the initialisation scheme explicitly.
2) Even though the authors have managed to induce “grokking” for MNIST and other tasks, the exercise seemed very artificial: yes, we can induce grokking by chopping off most of the dataset and raising the initialisation range. However, grokking does not seem to be something that we necessarily want to induce: looking at the results obtained, avoiding overfitting in the first place seems to still be the better option. Is the significance of this phenomenon perhaps blown out of proportion? I wish the relevance of grokking to standard ML tasks in light of the new experiments was discussed a bit more thoroughly.
3) The paper has minor language mistakes:
Page 5: “For large data size say the full dataset” -> e.g. the full dataset
Page 6: “by constrast” -> contrast
Page 9: “dramaticness” -> not a valid word, rather use “severity”

**Summary Of The Paper:**

The paper analyses the phenomenon of grokking (discovering solutions that generalise well after the model has overfit the data) from the perspective of loss landscapes, specifically the disparity between the training and the generalisation landscapes. The authors show that grokking can be induced on standard ML benchmarks such as MNIST classification by (1) increasing the weight initialisation range, (2) drastically reducing the number of data points in the training dataset. Further, the authors conduct experiments to show that grokking is linked to the correlation between the discovery of a good representation and generalisation. For loss landscapes analysis, low-dimensional projections are used, where the weights are represented as a weight norm. The reduced 1D loss landscape visualisation seems to be a powerful tool.

**Summary Of The Review:**

Overall, this is a good paper with brilliant loss landscape visualisations. Please refer to the least of strengths and weaknesses for the justification of my recommendation.

---

> ### Author Response · Authors · 2022-11-14
> **Response to ENpU**
>
> Dear Reviewer ENpU, thank you so much for the review and the constructive suggestions! Here is our response to your questions and concerns.
>
> Q1: The authors refer to “standard initialisation”, and then explain that it stands for default initialisation in PyTorch. It’s better to define them in explicit terms.
>
> A1: Agree, thanks for pointing out this! We have added a footnote on page 2 that “For linear layers, each weight $w\sim U[-\sigma,\sigma]$ and bias $b\sim U[-\sigma,\sigma]$ where $\sigma = 1/\sqrt{\rm fan\\_in}$, and $U[a,b]$ denotes uniform distribution on $[a,b]$.”.
>
> Q2: Grokking does not seem to be something that we necessarily want to induce: looking at the results obtained, avoiding overfitting in the first place seems to still be the better option. Is the significance of this phenomenon perhaps blown out of proportion? I wish the relevance of grokking to standard ML tasks in light of the new experiments was discussed a bit more thoroughly.
>
> A2: That’s a very good point! We feel the same - grokking (delayed generalization) is a pathological phenomenon that should be avoided rather than induced. Part of our motivation for inducing grokking is that if we can induce grokking (brings good to bad), maybe we are also able to eliminate grokking (brings bad to good).
>
> In our initial submission we did not have any experiment that aims to eliminate grokking, however, we later discovered that it is possible to (almost) eliminate grokking by carefully controlling the model weight norm. Specifically, we find that constraining the model at a smaller weight norm can nearly eliminate grokking on algorithmic datasets (see Appendix B and Figure 8). This hints that standard training strategies (standard initialization, no weight norm constraint) may not always be optimal. It would be an interesting direction to test whether our training tricks (constraining weight norm to be small) can help standard ML datasets generalize better and/or faster.
>
> Q3: The paper has minor language mistakes: Page 5: “For large data size say the full dataset” -> e.g. the full dataset Page 6: “by constrast” -> contrast Page 9: “dramaticness” -> not a valid word, rather use “severity”.
>
> A3: Thank you for reading so carefully! We have corrected them as suggested.

---

### Official Review · Reviewer_itfw · 2022-10-25

**Confidence:** 4
**Correctness:** 3
**Technical Novelty And Significance:** 3
**Empirical Novelty And Significance:** 3
**Recommendation:** 8

**Clarity, Quality, Novelty And Reproducibility:**

Clarity: The paper is written fairly clearly and the figures convey the main points. The experimental setup is unclear in Section 5.
Quality: The hypothesis is well-formed and the experiments support the hypothesis.
Novelty: To my knowledge, this paper is the first to show grokking for non-algorithmic datasets. While there have been some papers that explore some aspects of grokking, this paper adds to that literature.


**Strength And Weaknesses:**

Strengths:
1. The LU mechanism hypothesis is clearly stated, and the proposal to construct the landscape with fixed weight norm solutions is a nice idea to verify it (barring the possible optimization concerns for the constrained optimization problem).
2. The experiments on MNIST, IMDB support the LU mechanism hypothesis - I particularly liked Figure 3 as it illustrates the idea clearly.
3. The authors are able to demonstrate grokking for non-algorithmic datasets by changing the sample size and weight initialization - this is an important finding.
4. The proposed difference between algorithmic and non-algorithmic datasets is plausible - in particular that the weight norm increases and then drops as in Figure 9. The representation messiness argument also supports this conclusion, but I found this to be a little weak (see comments below).

Weaknesses:
1. Representation messiness:
- The experimental setup is hard to understand in Section 5.1 - I would suggest the authors include an appendix section describing the exact architecture, and the trainable/frozen components of this architecture. Does R_random depend on the input at all? If not, how is this a good proxy for representation learning?
- This part of the paper makes a reduction to a different architecture (from Liu et. al. 2022) and assumes that the representations are a convex combination of linearly separable and random gaussian representations. It is not clear to me why this is general and if the conclusions transfer to a setting where the representations are learnable (as is the case in grokking generally).
2. Minor complaints:
- The authors claim that changing the x-axis to weight norm removes double descent - this is only true for some limited cases and does not hold generally for neural networks as is also stated in the reference cited by authors. I would recommend that the authors add this clarification.
- Figure 6: Please state which experimental setup these loss landscapes have been plotted for.


**Summary Of The Paper:**

This paper proposes the LU mechanism for the phenomenon of Grokking observed by Power et. al. 2020 - which states that the train and test curves follow an L and U shaped curve with weight norms respectively. They verify this observation in a student teacher setup, and show that it can arise in non-algorithmic datasets if initialized in a certain weight regime for appropriate sample size. Further, they try to differentiate the algorihtmic and non-algorithmic datasets in terms of their dependence on representations.

**Summary Of The Review:**

The authors state three conclusions in their paper - the LU mechanism, the occurrence of grokking for wide range of datasets under certain conditions and the dependence of grokking on learning representations. The first two are well-supported by their experiments as the setup is general enough that the conclusions are believable. However, the third conclusion is made in an artificially reduced setup. While this setup supports the hypothesis, it is unclear to what extent this is related to the general phenomenon of grokking where representations are learnt and not fixed. Nevertheless, the contribution of the first two conclusions and weak evidence for the third conclusion seems enough for acceptance to the conference. I would recommend the authors include some discussion for their choice of setup in Section 5.

---

> ### Author Response · Authors · 2022-11-14
> **Response to itfw**
>
> Dear Reviewer itfw, thank you very much for the review and constructive feedback! Here is our response to the questions and concerns.
>
> Q1: The experimental setup is hard to understand in Section 5.1 - I would suggest the authors include an appendix section describing the exact architecture,  and the trainable/frozen components of this architecture.
>
> A1: We apologize for the confusion about our setups. As suggested, we have added Appendix D that explicitly distinguishes three setups - landscape analysis, reduced trajectory analysis and full trajectory analysis.
> * Summary of Appendix D here: The landscape analysis is related to the LU mechanism, and the full trajectory analysis is the normal grokking setup where both representation and decoder are trainable. Since the reduced trajectory analysis only appears in Section 5.1, we agree that it can easily get confused with the other two. The illustrative path A -> E in Figure 6 is a result of the reduced trajectory analysis, where the representation is trainable, but only in 1D (only allowed to move between the initial random Gaussian and final linear representation). We agree that this 1D assumption may be not very realistic, yet it agrees with the normal grokking setup in that: the weight norm first increases and then decreases, and the decrease in weight norm is highly correlated with generalization, as we show in Appendix B.
>
> Q2: The authors claim that changing the x-axis to weight norm removes double descent - this is only true for some limited cases and does not hold generally for neural networks as is also stated in the reference cited by authors. I would recommend that the authors add this clarification.
>
> A2: We agree and appreciate it! We’ve added “at least for linear regression” to make it less overclaiming.

---

### Official Review · Reviewer_ceW8 · 2022-10-26

**Confidence:** 3
**Correctness:** 3
**Technical Novelty And Significance:** 3
**Empirical Novelty And Significance:** 3
**Recommendation:** 8

**Clarity, Quality, Novelty And Reproducibility:**

Clarity: This paper is well-written and easy to follow.

Quality: High, all claims in this paper are well supported by thorough analysis and empirical experiments.

Novelty: It is novel to understand grokking from the loss landscapes. The experiment results are impressive and interesting.

Reproducibility: Hard. As claimed in the paper, grokking itself is hard to observe and depends on many factors.


**Strength And Weaknesses:**

Strength:

(1) It is interesting and novel to understand grokking from the lens of neural loss landscapes. More importantly, the connection between grokking and data size, weight decay, and representation learning are demonstrated with sufficient empirical analysis.

(2) It conducts sufficient and intuitive experiments to reveal the dependence of grokking on representation learning, which is helpful for the community to understand grokking in the learning process with complex datasets. Meanwhile, it contributes a useful new tool for characterizing data-model interaction and representation learning.

Weaknesses:

(1) Some terms should be explained in the paper for self-consistency, such as algorithmic datasets.

(2) The color bar of Fig.6 (b) is in [0.4, 2.0], which is different from other figures. And as claimed in the paper, the error = 1 - acc. This is inconsistent.

**Summary Of The Paper:**

This paper studies the properties of grokking from the loss landscapes. The phenomenon of "LU" mechanism of the training and test loss is adopted to understand grokking from its dependence on data size, weight decay, and representations. In particular, the experiment tailor-designed to reveal the connection between grokking and deep representation learning is interesting and impressive.

**Summary Of The Review:**

Overall, it is a decent paper.  it proposes to interpret the grokking phenomenon from the perspective of loss landscapes. And sufficient empirical analysis is provided to support the claims of the paper.

---

> ### Author Response · Authors · 2022-11-14
> **Response to ceW8**
>
> Dear Reviewer ceW8, thank you for the review and helpful suggestions! Here is our response to your questions:
>
> Q1: Some terms should be explained in the paper for self-consistency, such as algorithmic datasets.
>
> A1: Thanks for pointing out this! We have added descriptions at the beginning of Section 5.1: “Algorithmic datasets concern with learning a binary operation $a\circ b=c$ ($a,b,c$ are symbols) with neural networks, which aim to predict $c$ from input $(a,b)$.”
>
> Q2: The color bar of Fig.6 (b) is in [0.4, 2.0], which is different from other figures. And as claimed in the paper, the error = 1 - acc. This is inconsistent.
>
> A2: We apologize for the confusion. We used either loss and error=1-acc in the paper. For Figure 6, the error landscape looks a bit noisy (since the dataset is small), so we plotted loss instead. We now clarify below Eq. (1) that “In this paper, we set $f$ as train/test loss/error, but it also applies to other metrics of interest.”

---

### Author Response · Authors · 2022-11-14
**Summary of changes**

Firstly, we want to thank all reviewers for their high-quality reviews and constructive feedback. This first round of reviews prompted us to make a number of improvements to our manuscript. The revised version not only contains improvements in clarity, but also adds extra experiments that we believe make our conclusion stronger and more general. In the updated manuscript, major revisions, as well as passages relevant to our answers, are highlighted in blue. Here is a summary of our revisions/additions:

**1. Experiment on eliminating grokking**
* In Appendix B (Figure 8b), we now show that it is possible to (almost) eliminate grokking on algorithmic datasets, by initializing and constraining the model weight to a smaller norm.
* Reviewer ENpU was concerned that inducing grokking may not have practical implications for standard tasks. Now that we show we can eliminate grokking (speed up generalization) with special training tricks, these tricks may have the potential to improve generalization for standard tasks, an interesting direction for future research.
* Reviewer wgE4 expressed concern that the LU mechanism may not be the cause of grokking, that weight norm may just be correlated with some other, more fundamental property of the models. We think that our ability to almost eliminate grokking by constraining weight norm during training strengthens our theory. While the model weight norm and LU mechanism picture we’ve presented may not be the only cause of grokking, our ability to somewhat control grokking by constraining the weight norm suggests that it is a core aspect of the story of grokking.

**2. Experiment on the relation between generalization time and weight decay**
* In response to Reviewer wgE4’s concern that “the teacher-student setup may not be transferable more broadly”, we test the relation “generalization time is inversely proportional to weight decay” in both algorithmic datasets and MNIST, finding the relation holds approximately true. Results are included in Appendix C.

**3. Clarifying setups of Section 5.1**
* In response to Reviewer itfw’s suggestion, we added Appendix D to describe details of our toy algorithmic setups in Section 5.1.

---

### Decision · Program_Chairs · 2023-01-20

**Decision:**

Accept: notable-top-25%

**Justification For Why Not Higher Score:**

Because they use MSE loss in their experiment and it is not clear if the results extend to using cross-entropy

**Justification For Why Not Lower Score:**

Because the paper does a nice job of explaining a mechanism that leads to grokking

**Metareview: Summary, Strengths And Weaknesses:**

This paper studies the grokking phenomenon and provide an explanation related to the norm of the weights. In particular, the hypothesis that when the weights are initialized to have higher norm than the final solution, then there will be an initial slow period when the norm of the weights increases followed up by a fast period where the performance improves significantly. Authors also show that using their explanation they can produce grokking in variety of settings. Perhaps the main weakness of the work is that main experiments are done with MSE loss. Given the sensitivity of MSE loss to the norm of the weights, it is not clear if similar results can be shown for cross-entropy. I think this paper shows that grokking can happen because of the norm of the weights but not that all observed grokking phenomena are because of the norm of the weights.

**Note From Pc:**

if the above contains the word "oral" or "spotlight" please see: "oral" presentation means -> notable-top-5% and "spotlight" means -> notable-top-25%. As stated in our emails, we are disassociating presentation type from AC recommendations